# Pyroptosis in Cancer: Friend or Foe?

**DOI:** 10.3390/cancers13143620

**Published:** 2021-07-20

**Authors:** Xiuxia Lu, Tianhui Guo, Xing Zhang

**Affiliations:** Melanoma and Sarcoma Medical Oncology Unit, State Key Laboratory of Oncology in South China, Collaborative Innovation Center for Cancer Medicine, Sun Yat-sen University Cancer Center, Guangzhou 510060, China; luxx@sysucc.org.cn (X.L.); guoth@sysucc.org.cn (T.G.)

**Keywords:** pyroptosis, gasdermin, inflammasome, cancer, immunity

## Abstract

**Simple Summary:**

Pyroptosis is a new form of programmed cell death that differs from apoptosis in terms of its release of inflammatory factors and its characteristic bubble-like morphology. Pyroptosis was first discovered in the process of immune defense against bacterial infection, but the field of research soon spread to other inflammatory diseases and cancer. As cancer constitutes a serious risk for public health, numerous studies investigating pyroptosis in cancer have been carried out during these years. Tumorigenesis and new therapeutic treatments have been the focus of much recent research. This review discusses the role of pyroptosis in tumorigenesis and its influence on tumor immunity.

**Abstract:**

Pyroptosis is an inflammatory form of programmed cell death that is mediated by pore-forming proteins such as the gasdermin family (GSDMs), including GSDMA-E. Upon cleavage by activated caspases or granzyme proteases, the N-terminal of GSDMs oligomerizes in membranes to form pores, resulting in pyroptosis. Though all the gasdermin proteins have been studied in cancer, the role of pyroptosis in cancer remains mysterious, with conflicting findings. Numerous studies have shown that various stimuli, such as pathogen-associated molecular patterns (PAMPs), damage-associated molecular patterns (DAMPs), and chemotherapeutic drugs, could trigger pyroptosis when the cells express GSDMs. However, it is not clear whether pyroptosis in cancer induced by chemotherapeutic drugs or CAR T cell therapy is beneficial or harmful for anti-tumor immunity. This review discusses the discovery of pyroptosis as well as its role in inflammatory diseases and cancer, with an emphasis on tumor immunity.

## 1. Introduction

Programmed cell death mediated by specific signaling pathways plays a vital role in morphogenesis, the maintenance of homeostasis, and various diseases [1]. According to the morphology changes involved, programmed cell death can be divided into lytic and non-lytic cell death. Apoptosis is a well-studied form of non-lytic cell death without the release of pro-inflammatory factors. However, pyroptosis and necroptosis are lytic and highly inflammatory in nature [2,3]. Cross talk between the various forms of cell death is mediated by numerous pathways. Among these, caspase 8 is considered as a molecular switch that controls apoptosis, necroptosis, and pyroptosis [4,5].

Pyroptosis was first studied in the field of immune defense against pathogens. This field of research gradually extended to other diseases, especially cancer, as cancer has become a leading heath menace in recent decades. Many studies have found that tumor cells can be killed through pyroptosis, but the role of pyroptosis in tumorigenesis or anti-tumor immunity has not yet been clearly summarized. In this review, we aim to discuss how pyroptosis was discovered and the roles of pyroptosis in human diseases, including tumors, with an emphasis on the relationship between pyroptosis and tumor immunity.

## 2. The Discovery of Pyroptosis

Pyroptosis was first observed in 1992 when A. Zychlinsky treated macrophages with Shigella flexneri and saw lytic cell death [6]. However, this was considered to be a case of apoptosis at that time, and not until 2000 did Boise and Collins name it pyroptosis [7]. Cookson [8] demonstrated that, unlike apoptosis, pyroptosis was associated with the release of some proinflammatory factors and was dependent on caspase-1 instead of caspase-3, as in apoptosis. With the breakthrough in research on the inflammasome [9], pyroptosis was found to be induced upon the activation of caspase-1/4/5/11 and to occur in both macrophage and non-macrophage cells [10,11,12,13,14]. The underlying mechanism of how inflammatory caspases trigger pyroptosis remained unknown until Feng Shao [15] discovered that one of the gasdermin proteins, GSDMD, is required for caspase-1-mediated pyroptosis. When cells respond to an exogenous pathogen or endogenous damage, a canonical inflammasome is formed by the adaptor protein ASC and upstream sensors NLRP1b/NLRP3/NLRC4/AIM2/Pyrin, which in turn activates the effector pro-caspase 1, leading to its self-cleavage. The activated caspase 1 then cleaves GSDMD and the inflammatory factors pro-IL-18 and IL-1β. In the non-canonical inflammasome pathway, lipopolysaccharide (LPS) directly binds to and activates pro-caspase 4/5 or murine pro-caspase 11. Then, the activated caspases cleave GSDMD, contributing to the oligomerization of the N-terminal in membranes to form pores [16,17,18,19,20], leading to lytic cell death and the release of inflammatory cytokines such as IL-18, IL1β, and HMGB1 [21]. Later, it was discovered that the gasdermin family GSDMA/B/C/D/E could all be cleaved by activated caspases and granzyme proteases, and the N-terminal oligomerizes in membranes to form pores [22], leading to pyroptosis (Figure 1).

## 3. The Role of Pyroptosis in Inflammatory Diseases

Although pyroptosis was first discovered in Shigella flexneri-infected macrophages [23], Salmonella, various other bacteria [24], and viruses soon demonstrated pyroptosis inducing capability in target cells besides macrophages. In addition to its role in inflammatory diseases, pyroptosis also plays a vital part in other ailments. Diabetes is a major risk factor underlying cardiovascular diseases. Studies have shown that NLRP3 is a mediator of pyroptosis in cardiomyocytes, leading to diabetic cardiomyopathy [25,26,27]. Moreover, nicotine [28], cholesterol crystals [29], oxLDL [30], and cadmium [31] can disrupt the coronary arterial endothelial cells via pyroptosis, causing atherosclerosis. Additionally, pyroptosis in vascular smooth muscle cells and monocytes/macrophages also accelerates the pathogenesis of coronary atherosclerosis. Pyroptosis has also been reported in common neurological disorders. Increased intracellular calcium, mitochondrial dysfunction, and other endogenous compounds can activate the NLRP3 inflammasome and trigger pyroptosis in microglia and dopamine neurons, increasing the incidence of neuroinflammation and promoting the progression of Parkinson’s disease [32] and inflammatory demyelination during multiple sclerosis (MS) [33].

Taken together, cell death by pyroptosis has been found to be a common factor associated with the progression of many inflammation-associated diseases. Numerous studies have shown that it is possible to attenuate inflammation by blocking pyroptosis with inhibitors targeting caspase-1/3/4/5/11 or the NLRP3 inflammasome. HIV infection induces the cell death of quiescent lymphoid CD4 T cells by caspase-1-mediated pyroptosis, and those dying CD4 T cells release inflammatory signals that cause more cells to die [34]. This indicates that caspase 1 inhibitors could be a new class of ‘anti-AIDS’ therapeutic targets.

## 4. The Roles of Pyroptosis in Cancer: Promoting or Inhibiting the Tumor?

### 4.1. The Role of Inflammasomes in Cancer

Inflammasomes are protein complexes that are assembled in response to pathogens or intercellular danger. The type of inflammasome involved depends on the sensor used. These sensors consist of pyrin, AIM2, and nucleotide-binding domain (NOD)-like receptors (NLRs) such as NLRP1, NLRP3, and NLRC4, among which NLRP3 has been the most widely studied [35]. The NLRP3 inflammasome is involved in a variety of inflammation-associated diseases. In terms of tumors, the NLRP3 inflammasome could enhance the proliferation and migration of A549 lung cancer cells [36] (Table 1), and an increased expression of NLRP3 was positively correlated to tumor growth in oral squamous cell carcinoma (OSCC) [37] (Table 1). Another study showed that NLRP3 could promote epithelial to mesenchymal transition in colon cancer cells in an inflammasome-independent manner [38] (Table 1). Moreover, the NLRP3 inflammasome is involved in chemoresistance in oral squamous cell carcinoma [39] (Table 1) and insensitivity to radiotherapy in glioblastoma [40] (Table 1). Various studies have confirmed that the NLRP3-mediated release of IL-18 [41,42,43] or IL-1β [44,45,46,47] enhances the malignancy of lymphoma, gastric cancer, breast cancer, colon cancer, and glioblastoma via promoting the proliferation of cancer, immunosuppression, angiogenesis, and metastasis [48,49]. Similarly, Zhai et al. studied the tumor-promoting role of NLRP1 and reported that NLRP1 could augment the activation of inflammasomes and suppress apoptosis in metastatic melanoma [50]. 

On the contrary, a number of studies have also shown the anti-tumor activity of inflammasomes. Lower levels of NLRP3 [51] (Table 1), NLRP1 [54], and AIM2 [55] proteins were found in hepatocellular carcinoma (HCC), colorectal cancer (CRC), and gastric cancer (GC) compared to the adjacent normal tissue. A low level of these proteins was correlated with the advanced stage and poor prognosis of HCC, suggesting that the inflammasomes served as negative regulators in HCC tumorigenesis. The anti-tumor role of inflammasomes has been extensively studied in colitis-associated cancer, showing that the secretion of IL-18 mediated by the NLRP3 inflammasome could protect enterocytes from early-stage colitis induced by dextran sulfate sodium (DSS) or azoxymethane (AOM) [52,56,57,58] (Table 1). Furthermore, a study found that NLRC4 in tumor-associated macrophages (TAMs) inhibited the progression of melanoma independently of inflammasome activation [59]. Additionally, the NLRP3 inflammasome in Kupffer cells suppressed colorectal cancer metastatic growth in the liver by promoting the maturation of NK cells and tumoricidal activity mediated by IL-18 [53] (Table 1).

Overall, the role of the inflammasome in different cancer types is ambiguous (Figure 2).

### 4.2. The Role of Gasdermin Family Proteins in Cancer

Gasdermin family proteins, which include GSDMA, GSDMB, GSDMC, GSDMD, and GSDME, had been studied in other fields but not in cancers. Only in the last few years have their roles in pyroptosis and cancer been studied.

#### 4.2.1. GSDMA, GSDMB and GSDMC 

To date, the roles of GSDMA/B/C in tumors have been described only vaguely, and very limited information is available about them. GSDMA is usually expressed in epithelial cells and its lineage GSDMA3 has been proposed to be involved in epidermal differentiation [60], TNF-α-induced apoptosis [61], skin inflammation, and hair loss [62,63,64]. Polymorphisms of GSDMB as well as GSDMA were found to be linked to the development of childhood asthma [65,66,67]. GSDMA is frequently silenced in gastric cancers [68,69], indicating that GSDMA might function as a regulator of tumor suppression. Interestingly, GSDMB could be considered as an oncogene because of its enhanced expression in gastric cancers, hepatic carcinomas, cervical tumors, and breast cancer [70,71,72]. Additionally, many studies have also proposed that GSDMB possessed protumor functions such as migration, metastasis, and resistance to therapy in HER2 breast cancer [73,74].

Due to its constricted expression, there are no known diseases associated with GSDMC. GSDMC has been reported to be upregulated by transforming growth factor beta receptor 2 (TGFBR2) mutation in colorectal cancer (CRC), promoting tumor cell proliferation [75]. Likewise, the overexpression of GSDMC was confirmed in lung adenocarcinoma (LUAD) patients, acting as a promising predictive factor [76]. A recent study demonstrated that, under hypoxia, nuclear PD-L1 was found to enhance the expression of GSDMC, and breast cancer patients with an increased expression of GSDMC had poorer outcomes, suggesting that the chronic tumor necrosis induced by GSDMC in the center of hypoxia regions may promote tumor progression [77]. However, GSDMC expression is suppressed in some cell lines of both esophageal and gastric cancers [78], supporting its potential role as an anti-tumor agent.

From the above studies, we could not decide the definite roles of GSDMA/B/C in cancer. However, as a substrate of caspases or granzymes, GSDMs could be cleaved and the oligomerization of N-terminals in membranes would trigger pyroptosis, which has been confirmed in many cancers.

#### 4.2.2. GSDMD

Despite its role in innate immunity, GSDMD also plays a crucial part in carcinogenesis. The protein level of GSDMD was positively associated with the aggressiveness of non-small cell lung cancer (NSCLC). GSDMD modulated the EGFR/Akt signaling in NSCLC and promoted the proliferation of tumor by inhibiting apoptosis [79]. However, another study showed that the expression of GSDMD in gastric cancer (GC) was decreased compared to that in matched adjacent non-cancerous tissues and that the downregulation of GSDMD accelerated S/G2 cell transition, suggesting that GSDMD may serve as a tumor suppressor [80].

Many small-molecule inhibitors or drugs displaying tumoricidal activity through apoptosis or autography were also found to function by inducing GSDMD-mediated pyroptosis. Docosahexaenoic acid (DHA), an omega-3 fatty acid with anticancer effects, was verified to induce pyroptotic cell death in triple-negative breast cancer cells [81] (Table 2). Likewise, α-NETA induces the pyroptosis of epithelial ovarian cancer cells through the GSDMD/caspase-4 pathway [82] (Table 2). Polyphyllin VI (PPVI) [83] (Table 2) suppressed the proliferation of non-small cell lung cancer (NSCLC) by inducing pyroptosis via the induction of the ROS/NF-κB/NLRP3/GSDMD signal axis in NSCLC. Metformin could induce GSDMD-mediated pyroptosis in esophageal squamous cell carcinoma (ESCC) by targeting the miR-497-PELP1 axis, indicating that pyroptosis-inducing reagents could serve as alternative treatments for chemo- and radiotherapy refractory ESCC [84] (Table 2). Many new therapeutic targets and tumor markers have also been discovered in the process of studying tumoricidal drugs and pyroptosis. Small-molecule inhibitors of the serine dipeptidases DPP8 and DPP9 (DPP8/9) activated CARD8 and caspase-1 to trigger pyroptosis in the majority of human acute myeloid leukemia (AML) [85] (Table 2), indicating the protein CARD8 could be a novel therapeutic target for AML. The activation of pyroptosis in head and neck squamous cell carcinoma (HNSCC) was found to be a calcium-dependent process, and the reduced expression of calcium ion regulator CD38 could prevent inflammasome-induced pyroptosis in HNSCC, indicating that CD38 may function as a tumor suppressor in HNSCC [86].

Overall, despite its uncertain role in tumorigenesis, GSDMD-mediated pyroptosis in cancer has attracted huge interest from researchers worldwide and has remained a critical study topic, indicating its importance as a new therapeutic target.

#### 4.2.3. GSDME

As one of the members of the gasdermin family, GSDME is also known as deafness, autosomal dominant 5 (DFNA5). Based on several studies, the mutations at the DNA level result in the skipping of exon 8 [93,94,95] and the truncation of the protein, which augments the intrinsic pore-forming activity due to loss of the inhibitory C-terminal domain and thus damages hearing-associated cells and leads to hearing loss [96,97,98]. In contrast, the inactivation of GSDME in the form of decreased expression instead of genetic mutation is associated with cancer. Methylation analysis shows that in breast cancer [99,100,101,102] and colorectal cancer [103,104,105], the hypermethylated promoter of GSDME was relevant to patients’ prognosis, suggesting that GSDME methylation could be a potential detection and prognostic marker [99,105]. In most cancers, GSDME expression is decreased in tumor tissues compared to normal tissues [99,102,104,106,107]. Furthermore more, the induced expression of GSDME in cancer cell lines leads to decreased proliferation [108] and suppressed colony formation [104], while GSDME deficiency contributes to the malignancy of tumors [109] and drug resistance [110]. These studies have suggested that GSDME may function as a tumor suppressor gene. However, some other studies found that there is no clear correlation between GSDME promoter hypermethylation and GSDME expression [107], and there is no significant difference in GSDME expression between tumors and normal tissues [103]. Furthermore, GSDME-WT cancer cells develop tumors of comparable size to that of the GSDME-KO cells in colon cancer [87,111] (Table 2) and lung cancer [112] models, demonstrating that GSDME may not be involved in tumorigenesis. Surprisingly, one study even reported that higher GSDME expression in esophageal squamous cell carcinoma [88] (Table 2) was considered as a promising prognostic marker. Based on the above studies, the roles of GSDME in cancer are still controversial.

After Feng Shao discovered that the activation of caspase-3 by chemotherapeutic agents could trigger pyroptosis by cleaving GSDME [113], numerous studies showed that small-molecule inhibitors or traditional chemotherapeutic drugs could cause cell death via pyroptosis when tumor-expressing GSDME was cleaved by activated caspase-3 in various cancers [87,88,89,92] (Table 2). As we mentioned before that the expression of GSDME was associated with the sensitivity of chemotherapy, studies showed that GSDME deficiency could impair the efficacy of cisplatin [89] (Table 2), sulfasalazine together with iron dextran [114], or ceritinib [112]. However, in some other cases, GSDME did not affect the tumoricidal activity of 5-FU [90] (Table 2) or lobaplatin [87] (Table 2). All these tumoricidal substances kill cancer cells not only by pyroptosis but also through apoptosis or autophagy. However, in the absence of GSDME, pyroptosis could not be triggered.

Taken together, the role of GSDME in cancer could not be determined, with conflicting results in tumorigenesis. Thus, it still necessary to conduct further research, especially concerning the induction of pyroptosis in tumors.

#### 4.2.4. The Significance of Pyroptosis in Tumorigenesis or Tumor Progression?

Thus far, all gasdermin family proteins have been studied in cancers but neither the relationship between their expression and tumor malignancy nor the function of GSDMs in tumorigenesis are not consistent. Although some studies have shown that the ectopic expression of GSDMs promoted the formation of tumors [115], the underlying mechanism has not been revealed. It is well known that tumor development is a multi-stage process and different mechanisms operate at different times in various cancers [116,117]. The factors involved in tumor initiation, such as the loss-of-function mutations in p53 [118] or the formation of oncoprotein BCR-ABL [119], have been proven to play a role in the initial stages. As the tumor grows, the inner part of tumor cells suffers from more hypoxia stress, starvation, and other harsh conditions. One recent study found that pyroptosis in a small fraction of tumor cells in the central hypoxia region of cancer contributed to chronic tumor necrosis, which in turn suppressed the anti-tumor immunity and promoted tumor progression [77]. Without therapeutic regents, the pyroptosis of cancer cells triggered by an adverse tumor microenvironment is more likely to accelerate the progression. Thus, as far as we are concerned, pyroptosis is more consistently related to metastasis and late-stage solid tumors than to preneoplastic lesions or early-stage tumors. Moreover, gasdermin family proteins may have some other cellular functions despite their contribution to pyroptosis, and this also deserves further research.

## 5. The Correlation between Pyroptosis and Therapeutic Reagents

Apoptosis is the most widely studied process of tumor inhibition and numerous chemotherapeutic agents for inducing apoptosis have been extensively studied. Of late, many researchers have clearly demonstrated that apoptosis could be switched to pyroptosis [120]. Pyroptosis is quite common in cancer cells treated with traditional chemotherapeutic drugs or promising small-molecule targets. Though there is no solid evidence that GSDMs are linked to tumorigenesis, the expression of the gasdermin family members in caners determines whether pyroptosis could be induced or not. Moreover, the expression of GSDMs is always relatively higher in normal tissues as compared to cancers, and under this condition pyroptosis would be more easily induced in normal tissues instead of tumors, potentiating unwanted effects or side effects of tumoricidal drugs. One study showed that doxorubicin induced-cardiotoxicity was mediated by pyroptosis in cardiomyocyte regulated through the Bnip3-caspase-3-GSDME pathway [121]. Similarly, the chemotherapeutic drugs cisplatin and doxorubicin also induced nephrotoxicity through pyroptosis [122]. Based on these results, we could see that pyroptosis not only mediated the antitumor activity of the therapeutic agents but also their side effects. At present, studies in this area mostly focus on the therapeutic effects of the targeting drugs and whether the drugs induce pyroptosis and fuel anti-tumor immunity.

All in all, as a form of cell death, pyroptosis is universal among cancer cells and normal cells treated with tumoricidal reagents. Patients may benefit more from more tumor-specific treatment, such as the selective induction of pyroptosis in tumor cells, which directly kills tumors but also activates anti-tumor immunity.

## 6. The Relationship between Pyroptosis and Anti-Tumor Immunity

Numerous studies have revealed that pyroptosis could fuel anti-tumor immunity. The application of a biorthogonal system to GSDMA3 in tumor cells showed that pyroptosis may augment the antitumor immune response and increase the efficacy of immune checkpoint blockade [123] (Table 3). BRAF inhibitors together with MEK inhibitors could induce pyroptosis in melanoma with a higher HMGB1, more tumor-associated T cells, and less dendritic cell infiltration, while GSDME deficiency would reverse the anti-tumor immunity [91] (Table 2 and Table 3). Two other recent studies found that the granzymes released from CD8 +T cells and NK cells could cleave GSDMB/E and thus trigger the pyroptosis of tumor cells, indicating that pyroptosis might serve as an important effector in anti-tumor immunity [115,124] (Table 3). However, Gao Tan’s [21] study found that HMGB1, a proinflammatory factor released from GSDME-mediated pyroptotic epithelial cells, induces colorectal cancer proliferation through the ERK1/2 pathway. Furthermore, Liu’s research found that pyroptosis-released factors from tumor cells could again trigger pyroptosis in macrophages, resulting in the release of cytokines and subsequent cytokine release syndrome (CRS) [125] (Table 3). Moreover, as mentioned before, IL-18 has both pro [41,42,43] and antitumorigenic effects [52,53]. The effects of other intercellular substances in the tumor microenvironment, such as ATP, HMGB1, IL1β, and LDH, are still uncertain. This is a topic that deserves further study. From these results, we can see that pyroptosis mediates the tumoricidal effect of some inhibitors or cytolytic immune cells, and pyroptosis-released inflammatory factors may fuel the anti-tumor immunity or be harmful to patients by promoting tumor or causing inflammatory cascades. Thus, the relationship between pyroptosis and anti-tumor immunity is not certain and is worthy of further investigation.

## 7. Conclusions and Future Prospects

Compared with apoptosis or autophagy, pyroptosis is a form of proinflammatory cell death with released inflammatory factors. Pyroptosis was first discovered in the defense of pathogenic insults and later researchers found that pyroptosis mediates many inflammatory diseases. It has been proposed that the inhibition of pyroptosis through targeting caspase-1/3/4/5/11 or the NLRP3 inflammasome could relieve inflammatory diseases. However, some other studies have found that the induction of pyroptosis in tumor cells would enhance the tumoricidal effect. Various tumoricidal substances such as chemotherapeutic drugs or granzymes released from cytotoxic T cells could kill cancer cells via canonical or non-canonical pyroptosis. However, there is no confirmed conclusion that inflammasomes or GSDMs are involved in tumorigenesis or act as tumor suppressors, and it is not certain that pyroptosis induced by tumoricidal drugs is beneficial for tumor patients in the long run because other normal cells could also die by pyroptosis when stimulated by DAMPs released from pyroptotic tumor cells. Moreover, the underlying mechanism that regulates pyroptosis has not been revealed so far and the roles of pyroptosis in anti-tumor immunity are still ambiguous. Above all, every aspect of pyroptosis needs to be elucidated further by more in-depth studies to confirm its function and mechanism. Additionally, it would be also interesting to explore some drug candidates that could inhibit tumors via pyroptosis and facilitate anti-tumor immunity.

## Figures and Tables

**Figure 1 cancers-13-03620-f001:**
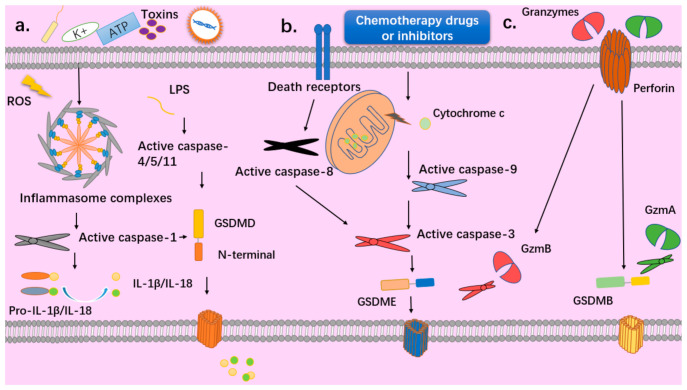
Different pathways to pyroptosis. (**a**) Canonical pyroptosis: PAMPs such as Gram-negative bacteria, viruses, toxins and DAMPs such as intracellular ROS, ATP, potassium, cadmium can activate the inflammasomes and activated caspase 1 can cleave GSDMD and pro-IL-1β/IL-18. Non-canonical pyroptosis: Bacterial lipopolysaccharide (LPS) directly binds to and activates pro-caspase 4/5 or murine pro-caspase 11, then the activated caspases cleave GSDMD. (**b**) Chemotherapeutic drugs or inhibitors can disrupt the mitochondrial membrane and the release of cytochrome c activates caspase 9 and caspase 3 to cleave GSDME and trigger pyroptosis. Caspase 3 could also be activated by caspase 8 when death receptors are stimulated. (**c**) Cytotoxic T lymphocytes (CTLs) and natural killer (NK) cells release perforin to deliver serine protease granzymes (Gzms) into target cells, and then GzmA and GzmB can cleave GSDMB/E, which triggers pyroptosis.

**Figure 2 cancers-13-03620-f002:**
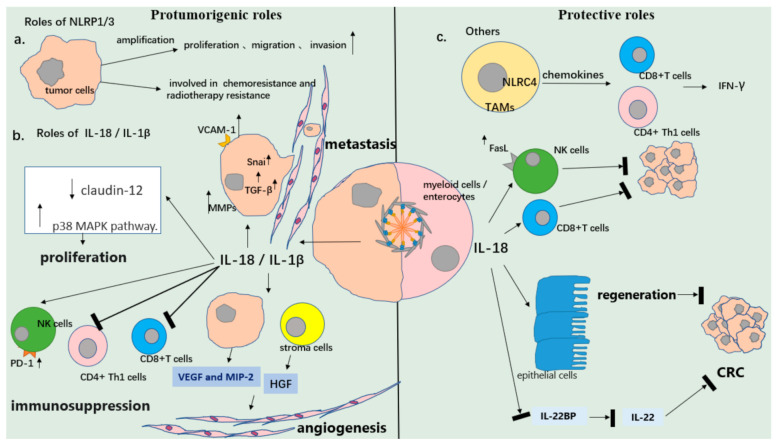
Diverse roles of the inflammasome in cancer. (**a**) NLRP3 mediates the proliferation and invasion of tumor cells. A higher expression of NLRP3 in tumor is associated with the clinical outcomes of patients. (**b**) The release of IL-18 /IL-1βfrom tumor cells induces tumor proliferation, immunosuppression, angiogenesis, and metastasis. However, IL-18 from myeloid cells or enterocytes can activate cytolytic immune cells and inhibit the formation of tumors (colorectal cancer) via initiating tissue repair. (**c**) Others. Inflammasomes such as NLRC4 in tumor-associated macrophages (TAMs) can inhibit tumor progression.

**Table 1 cancers-13-03620-t001:** Roles of the NLRP3 inflammasome in cancer.

Tumor Types	Significance	Experimental Models	Results	Refs.
Lung cancer	tumor promoter	LC cell lines	The activation of NLRP3 increased the tumor proliferation and migration.	[36]
OSCC	tumor promoter	OSCC cell lines and tissue	The enhanced expression of NLRP3 was correlated with tumor growth and metastasis.	[37]
			The upregulation of NLRP3 correlates with the chemoresistance of 5-FU in OSCC.	[39]
Colon cancer	tumor promoter	CRC cell lines	The NLRP3 level was increased during EMT, which was independent of inflammasome activation.	[38]
Glioma	tumor promoter	glioma cell lines	NLRP3 inflammasome contributed to the radiotherapy resistance in a xenograft mouse glioblastoma model.	[40]
Liver cancer	tumor suppressor	HCC tissue	The deficiency of the NLRP3 inflammasome was related to higher pathological grades and advanced clinical stages in HCC.	[51]
Colorectal cancer	tumor suppressor	Nlrp3−/−, Asc−/− and Caspase1−/− mice	NLRP3 inflammasome protected mice from colitis-associated colorectal tumorigenesis.	[52]
		macrophages surrounded CRC tissue	The NLRP3 inflammasome inhibited the growth of liver colon cancer metastatic tumor.	[53]

Abbreviations: LC, lung cancer; OSCC, oral squamous cell carcinoma; CRC, colorectal cancer; EMT, epithelial-mesenchymal transition; HCC, hepatocellular carcinoma.

**Table 2 cancers-13-03620-t002:** Induction of pyroptosis in cancer by various therapeutic reagents.

Therapeutic Reagents	Cancer Type	Pathway	Refs.
Docosahexaenoic acid (DHA)	Breast cancer	NF-κB/Caspase-1/GSDMD	[81]
α-NETA	Ovarian cancer	GSDMD/caspase-4	[82]
Polyphyllin VI (PPVI)	NSCLC	ROS/NF-κB/NLRP3/GSDMD	[83]
Metformin	ESCC	miR-497/PELP1/GSDMD	[84]
DPP8/9	AML	CARD8/Caspase-1/GSDMD	[85]
Cadmium	Breast cancer		[87]
BI2536 and Cisplatin	ESCC		[88]
Paclitaxel and cisplatin	Lung cancer	Caspase-3/GSDME	[89]
5-FU	Gastric cancer		[90]
BRAFi + MEKi	Melanoma		[91]
Doxorubicin	Melanoma	eEF-2K/GSDME	[92]

**Table 3 cancers-13-03620-t003:** Roles of gasdermin family members in tumor immunity.

Gasdermin Family Members	Roles in Tumor Immunity	Refs.
GSDMA3	A small fraction of pyroptotic tumor cells brought about the increased infiltration of cytotoxic T cells and CD4+ T helper cells and other immunological changes within the tumor.	[123]
GSDMB	Th expression of GSDMB wild type (WT) promoted cytotoxic T lymphocyte-mediated tumor clearance by antibodies to programmed cell death 1 (PD-1).	[124]
GSDME	The ectopic expression of GSDME inhibited tumor growth and enhanced tumor immunity with more and functional tumor-infiltrating natural-killer and CD8+ T lymphocytes.	[115]
GSDME	Treatment with BRAF inhibitors and MEK inhibitors in mouse melanoma promoted pyroptosis and the release of HMGB1, which enhanced the levels of tumor-associated T cells and damped dendritic cell infiltrates.	[91]
GSDME	Large amounts of damage-associated molecular pattern molecules (DAMPs) from pyroptotic cancer cells activate caspase 1 for GSDMD cleavage in macrophages, resulting in the release of proinflammatory cytokines and subsequent CRS.	[125]

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
