# Peer review of "Pyroptosis in Cancer: Friend or Foe?"

_cancers, 2021, doi:10.3390/cancers13143620_

Round 1

Reviewer 1 Report

General Comments

This review summarizes current knowledge of cell death by pyroptosis as a mechanism in cancer.  This is an important and timely topic.  The organization of the text is clear and appropriate.  Table 1 summarizing pyroptosis in response to treatment is a strength, since it simplifies understanding the discussion.  But there are concerns.

1. There are already a number of similar review papers on the topic of pyroptosis and cancer. Most reviews convey the expectation that all cancers can be “binned” together. In other papers, there is minimal or no consideration that there may be several different mechanisms operating at different times in various cancers. 

The contribution of this paper would be strengthened by considering cancer stage or grade or some other factor the authors might decide is appropriate, as described below.a. Is it possible that some of the conflicting findings are due to differences in tumor stage or grade, or other categories such as differences between animal model and human studies? Is pyroptosis more consistently related to metastasis and late-stage solid tumors than to preneoplastic lesions or early-stage tumors? 

  1. Similarly, are there studies suggesting pyroptosis is a more significant mechanism in treatment side effects (adverse events) or therapy resistance, than in tumor initiation and progression.
  2. Please clarify if the studies cited relate to immune cell pyroptosis or tumor cell pyroptosis (or if is it not known). This may be important since pyroptosis of tumor cells would release antigens and factors to stimulate immunity, while immune cell pyroptosis might abort anti-tumor immunity.  The authors may find the section on Pyroptosis and Anti-tumor immunity relevant, by Tang et al. (Ferroptosis, necroptosis, and pyroptosis in anticancer immunity. J Hematol Oncol 13, 110 (2020). https://doi.org/10.1186/s13045-020-00946-7). 
  3. According to the Introduction, the review “discusses the role of pyroptosis in tumorigenesis and its influence on the tumor microenvironment”. However, the final sections discuss tumor immunity and pyroptosis and but do not appear to significantly address the microenvironment (TmE).  The TmE is composed of multiple cells and factors (not just immune cells).  Is there a particular aspect of the TmE (e.g., immune cells vs non-immune cells) that is relevant to pyroptosis?  One suggestion is to modify the Introduction to a focus on tumor immunity rather than TmE.

Is pyroptosis important in cancer because it kills cells directly or because it activates tumor immunity (or both)?

Specific comments

Page 2 – section 2.   please define canonical vs noncanonical.  The terms are used without clarification.  A recent reference that clearly differentiates canonical and non-canonical may be useful. (cf Tan et al. Pyroptosis: a new paradigm of cell death for fighting against cancer. J Exp Clin Cancer Res 40, 153 (2021). https://doi.org/10.1186/s13046-021-01959)

Page 3 – Which type of inflammasome? There are multiple types as the authors mentioned elsewhere in the text.  The report of Zhou et al. may be relevant to the authors discussion (Zhou et al., Gasdermin E permits interleukin-1 beta release in distinct sublytic and pyroptotic phases. Cell Rep. 2021 Apr 13;35(2):108998. doi: 10.1016/j.celrep.2021.108998. PMID: 33852854; PMCID: PMC8106763). 

The manuscript would benefit from a professional editor.  Although the authors made their points, there were numerous grammar and spelling errors that sometimes made it hard to read. 

Author Response

Dear Editor:

We thank you very much for giving us an opportunity to revise our manuscript, we appreciate you very much for your positive and constructive comments and suggestions on our manuscript.

In order to minimize the writing mistakes, this article has been edited for proper English writing by a native English speaker. The changes will not influence the content and framework of the paper. The detailed response could be seen in the  attachment.

If you have any questions regarding this manuscript, please feel free to contact me. Looking forward to hearing from you.

Yours sincerely,

Lu Xiuxia, PhD,

Sun Yat-sen University Cancer Center

Reviewer 2 Report

In this manuscript, Lu et al. review the relatively novel field of pyroptosis in cancer. Pyroptosis might have opposing functions in cancer. In the tumor microenvironment, immune cells undergoing pyroptosis may contribute to an inflammatory environment that fuels the tumor, while pyroptosis in tumor cells may expose neoantigens and release DAMPs that together contribute to tumor immunity. The authors do a good job in reviewing these roles of pyroptosis, although they mainly focus on tumor immunity and may have included more insights from the side of the tumor microenvironment.

I only have a couple of remarks to improve this manuscript.

  1. Please have the text edited by a native English speaker. It is full of spelling and syntax errors.

  1. The section on ‘the role of pyroptosis in inflammatory diseases’ seems mostly redundant. It should be shortened or could even be omitted to allow more focused and in-depth discussions on the role of pyroptosis in cancer.

  1. Some additional tables in the manuscript could provide more detailed insights in the studies mentioned by the authors. For instance, references 46-59, 61, 65 and 69 all deal with the effect of the Nlrp3 inflammasome on cancer cells but very little information is provided. Are these human or mouse studies? In vitro or in vivo? Genetic studies or using pharmacological Nlrp3 inhibition? Or merely correlating observations with Nlrp3 expression or activity levels? Providing this info in a table would be very useful for the reader.

  1. Similarly, also a table listing experimental details on the tumor immunity findings obtained with the various Gasdermin family members could be useful for the reader.

  1. The authors refer to a large number of reviews that are outdated. Refs 1, 3, 7, 8, 9, 11 and 21 all are review articles dating from 2017 or earlier. Given the fast evolving pyroptosis field it would be better to refer the reader to fewer but more recent reviews.

Author Response

Dear Editor:

We thank you very much for giving us an opportunity to revise our manuscript, we appreciate you very much for their positive and constructive comments and suggestions on our manuscript.

We have studied reviewer’s comments carefully and have made revision which marked in red in the paper. We have tried our best to revise our manuscript according to the comments. 

In order to minimize the writing mistakes, this article has been edited for proper English writing by the a native English speaker with medical PhD.  The changes will not influence the content and framework of the paper.The detailed response could be seen in the attachment.

If you have any questions regarding this manuscript, please feel free to contact me. Looking forward to hearing from you.

Yours sincerely,

Lu Xiuxia  PhD,

Sun Yat-sen University Cancer Center

Round 2

Reviewer 1 Report

The changes in the revised manuscript appear to address the review concerns.  The addition of more summary tables was a positive improvement (although the number of words could be reduced for clarity).  The language editing was difficult to read since both old and new text ran together (better to cross out the old material).  Overall the content is improved but the language needs improvement.

Author Response

Dear Editor:

  Thank you very much for giving us positive and constructive comments.

   We have tried our best to revise our manuscript according to the comments. Attached please find the revised version, which we would like to submit for your kind consideration.

Yours sincerely,

Xiuxia Lu,  PhD,

Sun Yat-sen University Cancer Center
